# Low-Rank Time-Frequency Synthesis

**Cédric Févotte**
Laboratoire Lagrange
(CNRS, OCA & Université de Nice)
Nice, France
cfevotte@unice.fr

**Matthieu Kowalski**[*]
Laboratoire des Signaux et Systèmes
(CNRS, Supélec & Université Paris-Sud)
Gif-sur-Yvette, France
kowalski@lss.supelec.fr

## Abstract

Many single-channel signal decomposition techniques rely on a low-rank factorization of a time-frequency transform. In particular, nonnegative matrix factorization (NMF) of the spectrogram – the (power) magnitude of the short-time Fourier transform (STFT) – has been considered in many audio applications. In this setting, NMF with the Itakura-Saito divergence was shown to underly a generative Gaussian composite model (GCM) of the STFT, a step forward from more empirical approaches based on ad-hoc transform and divergence specifications. Still, the GCM is not yet a generative model of the raw signal itself, but only of its STFT. The work presented in this paper fills in this ultimate gap by proposing a novel signal synthesis model with low-rank time-frequency structure. In particular, our new approach opens doors to multi-resolution representations, that were not possible in the traditional NMF setting. We describe two expectation-maximization algorithms for estimation in the new model and report audio signal processing results with music decomposition and speech enhancement.

## 1 Introduction

Matrix factorization methods currently enjoy a large popularity in machine learning and signal processing. In the latter field, the input data is usually a time-frequency transform of some original time series $x(t)$. For example, in the audio setting, nonnegative matrix factorization (NMF) is commonly used to decompose magnitude or power spectrograms into elementary components [1]; the spectrogram, say $\mathbf{S}$, is approximately factorized into $\mathbf{WH}$, where $\mathbf{W}$ is the dictionary matrix collecting spectral patterns in its columns and $\mathbf{H}$ is the activation matrix. The approximate $\mathbf{WH}$ is generally of lower rank than $\mathbf{S}$, unless additional constraints are imposed on the factors.

NMF was originally designed in a deterministic setting [2]: a measure of fit between $\mathbf{S}$ and $\mathbf{WH}$ is minimized with respect to (w.r.t) $\mathbf{W}$ and $\mathbf{H}$. Choosing the "right" measure for a specific type of data and task is not straightforward. Furthermore, NMF-based spectral decompositions often arbitrarily discard phase information: only the magnitude of the complex-valued short-time Fourier transform (STFT) is considered. To remedy these limitations, a generative probabilistic latent factor model of the STFT was proposed in [3]. Denoting by $\{y_{fn}\}$ the complex-valued coefficients of the STFT of $x(t)$, where $f$ and $n$ index frequencies and time frames, respectively, the so-called Gaussian Composite Model (GCM) introduced in [3] writes simply

$$y_{fn} \sim N_c(0, [\mathbf{WH}]_{fn}), \tag{1}$$

where $N_c$ refers to the circular complex-valued normal distribution.[1] As shown by Eq. (1), in the GCM the STFT is assumed centered (reflecting an equivalent assumption in the time domain which

---

[*] Authorship based on alphabetical order to reflect an equal contribution.

[1] A random variable $x$ has distribution $N_c(x|\mu, \lambda) = (\pi\lambda)^{-1} \exp -(|x - \mu|^2/\lambda)$ if and only if its real and imaginary parts are independent and with distribution $N(\text{Re}(\mu), \lambda/2)$ and $N(\text{Im}(\mu), \lambda/2)$, respectively.

is valid for many signals such as audio signals) and its variance has a low-rank structure. Under these assumptions, the negative log-likelihood $-\log p(\mathbf{Y}|\mathbf{W}, \mathbf{H})$ of the STFT matrix $\mathbf{Y}$ and parameters $\mathbf{W}$ and $\mathbf{H}$ is equal, up to a constant, to the Itakura-Saito (IS) divergence $D_{\text{IS}}(\mathbf{S}|\mathbf{WH})$ between the power spectrogram $\mathbf{S} = |\mathbf{Y}|^2$ and $\mathbf{WH}$ [3].

The GCM is a step forward from traditional NMF approaches that fail to provide a valid generative model of the STFT itself – other approaches have only considered probabilistic models of the magnitude spectrogram under Poisson or multinomial assumptions, see [1] for a review. Still, the GCM is not yet a generative model of the raw signal $x(t)$ itself, but of its STFT. The work reported in this paper fills in this ultimate gap. It describes a novel signal synthesis model with low-rank time-frequency structure. Besides improved accuracy of representation thanks to modeling at lowest level, our new approach opens doors to multi-resolution representations, that were not possible in the traditional NMF setting. Because of the synthesis approach, we may represent the signal as a sum of layers with their own time resolution, and their own latent low-rank structure.

The paper is organized as follows. Section 2 introduces the new low-rank time-frequency synthesis (LRTFS) model. Section 3 addresses estimation in LRTFS. We present two maximum likelihood estimation approaches with companion EM algorithms. Section 4 describes how LRTFS can be adapted to multiple-resolution representations. Section 5 reports experiments with audio applications, namely music decomposition and speech enhancement. Section 6 concludes.

## 2 The LRTFS model

### 2.1 Generative model

The LRTFS model is defined by the following set of equations. For $t = 1, \ldots, T$, $f = 1, \ldots, F$, $n = 1, \ldots, N$:

$$x(t) = \sum_{fn} \alpha_{fn}\phi_{fn}(t) + e(t) \tag{2}$$

$$\alpha_{fn} \sim N_c(0, [\mathbf{WH}]_{fn}) \tag{3}$$

$$e(t) \sim N_c(0, \lambda) \tag{4}$$

For generality and simplicity of presentation, all the variables in Eq. (2) are assumed complex-valued. In the real case, the hermitian symmetry of the time-frequency (t-f) frame can be exploited: one only needs to consider the atoms relative to positive frequencies, generate the corresponding complex signal and then generate the real signal satisfying the hermitian symmetry on the coefficients. $\mathbf{W}$ and $\mathbf{H}$ are nonnegative matrices of dimensions $F \times K$ and $K \times N$, respectively.[2] For a fixed t-f point $(f, n)$, the signal $\boldsymbol{\phi}_{fn} = \{\phi_{fn}(t)\}_t$, referred to as *atom*, is the element of an arbitrary t-f basis, for example a Gabor frame (a collection of tapered oscillating functions with short temporal support). $e(t)$ is an identically and independently distributed (i.i.d) Gaussian residual term. The variables $\{\alpha_{fn}\}$ are *synthesis coefficients*, assumed conditionally independent. Loosely speaking, they are dual of the *analysis coefficients*, defined by $y_{fn} = \sum_t x(t)\phi_{fn}^*(t)$. The coefficients of the STFT can be interpreted as analysis coefficients obtained with a Gabor frame. The synthesis coefficients are assumed centered, ensuring that $x(t)$ has zero expectation as well. A low-rank latent structure is imposed on their variance. This is in contrast with the GCM introduced at Eq. (1), that instead imposes a low-rank structure on the variance of the analysis coefficients.

### 2.2 Relation to sparse Bayesian learning

Eq. (2) may be written in matrix form as

$$\mathbf{x} = \boldsymbol{\Phi}\boldsymbol{\alpha} + \mathbf{e} \,, \tag{5}$$

where $\mathbf{x}$ and $\mathbf{e}$ are column vectors of dimension $T$ with coefficients $x(t)$ and $e(t)$, respectively. Given an arbitrary mapping from $(f, n) \in \{1, \ldots, F\} \times \{1, \ldots, N\}$ to $m \in \{1, \ldots, M\}$, where $M = FN$, $\boldsymbol{\alpha}$ is a column vector of dimension $M$ with coefficients $\{\alpha_{fn}\}_{fn}$ and $\boldsymbol{\Phi}$ is a matrix of size $T \times M$ with columns $\{\boldsymbol{\phi}_{fn}\}_{fn}$. In the following we will sometimes slightly abuse notations by

indexing the coefficients of $\boldsymbol{\alpha}$ (and other variables) by either $m$ or $(f, n)$. It should be understood that $m$ and $(f, n)$ are in one-to-one correspondence and the notation should be clear from the context. Let us denote by $\mathbf{v}$ the column vector of dimension $M$ with coefficients $v_{fn} = [\mathbf{WH}]_{fn}$. Then, from Eq. (3), we may write that the prior distribution for $\boldsymbol{\alpha}$ is

$$p(\boldsymbol{\alpha}|\mathbf{v}) = N_c(\boldsymbol{\alpha}|\mathbf{0}, \text{diag}(\mathbf{v})) \,. \tag{6}$$

Ignoring the low-rank constraint, Eqs. (5)-(6) resemble sparse Bayesian learning (SBL), as introduced in [4, 5], where it is shown that marginal likelihood estimation of the variance induces sparse solutions of $\mathbf{v}$ and thus $\boldsymbol{\alpha}$. The essential difference between our model and SBL is that the coefficients are no longer unstructured in LRTFS. Indeed, in SBL, each coefficient $\alpha_m$ has a free variance parameter $v_m$. This property is fundamental to the sparsity-inducing effect of SBL [4]. In contrast, in LRTFS, the variances are now tied together and such that $v_m = v_{fn} = [\mathbf{WH}]_{fn}$ .

### 2.3 Latent components reconstruction

As its name suggests, the GCM described by Eq. (1) is a *composite* model, in the following sense. We may introduce independent complex-valued latent components $y_{kfn} \sim N_c(0, w_{fk}h_{kn})$ and write $y_{fn} = \sum_{k=1}^{K} y_{kfn}$. Marginalizing the components from this simple Gaussian additive model leads to Eq. (1). In this perspective, the GCM implicitly assumes the data STFT $\mathbf{Y}$ to be a sum of elementary STFT components $\mathbf{Y}_k = \{y_{kfn}\}_{fn}$ . In the GCM, the components can be reconstructed after estimation of $\mathbf{W}$ and $\mathbf{H}$ , using any statistical estimator. In particular, the minimum mean square estimator (MMSE), given by the posterior mean, reduces to so-called Wiener filtering:

$$\hat{y}_{kfn} = \frac{w_{fk}h_{kn}}{[\mathbf{WH}]_{fn}} y_{fn}. \tag{7}$$

The components may then be STFT-inversed to obtain temporal reconstructions that form the output of the overall signal decomposition approach.

Of course, the same principle applies to LRTFS. The synthesis coefficients $\alpha_{fn}$ may equally be written as a sum of latent components, such that $\alpha_{fn} = \sum_k \alpha_{kfn}$, with $\alpha_{kfn} \sim N_c(0, w_{fk}h_{kn})$. Denoting by $\boldsymbol{\alpha}_k$ the column vector of dimension $M$ with coefficients $\{\alpha_{kfn}\}_{fn}$, Eq. (5) may be written as

$$\mathbf{x} = \sum_k \boldsymbol{\Phi}\boldsymbol{\alpha}_k + \mathbf{e} = \sum_k \mathbf{c}_k + \mathbf{e} \,, \tag{8}$$

where $\mathbf{c}_k = \boldsymbol{\Phi}\boldsymbol{\alpha}_k$. The component $\mathbf{c}_k$ is the "temporal expression" of spectral pattern $\mathbf{w}_k$, the $k^{th}$ column of $\mathbf{W}$. Given estimates of $\mathbf{W}$ and $\mathbf{H}$, the components may be reconstructed in various way. The equivalent of the Wiener filtering approach used traditionally with the GCM would consist in computing $\hat{\mathbf{c}}_k^{\text{MMSE}} = \boldsymbol{\Phi}\hat{\boldsymbol{\alpha}}_k^{\text{MMSE}}$, with $\hat{\boldsymbol{\alpha}}_k^{\text{MMSE}} = \mathbb{E}\{\boldsymbol{\alpha}_k|\mathbf{x}, \mathbf{W}, \mathbf{H}\}$. Though the expression of $\hat{\boldsymbol{\alpha}}_k^{\text{MMSE}}$ is available in closed form it requires the inversion of a too large matrix, of dimensions $T \times T$ (see also Section 3.2). We will instead use $\hat{\mathbf{c}}_k = \boldsymbol{\Phi}\hat{\boldsymbol{\alpha}}_k$ with $\hat{\boldsymbol{\alpha}}_k = \mathbb{E}\{\boldsymbol{\alpha}_k|\hat{\boldsymbol{\alpha}}, \mathbf{W}, \mathbf{H}\}$, where $\hat{\boldsymbol{\alpha}}$ is the available estimate of $\boldsymbol{\alpha}$. In this case, the coefficients of $\hat{\boldsymbol{\alpha}}_k$ are given by

$$\hat{\alpha}_{kfn} = \frac{w_{fk}h_{kn}}{[\mathbf{WH}]_{fn}} \hat{\alpha}_{fn}. \tag{9}$$

## 3 Estimation in LRTFS

We now consider two approaches to estimation of $\mathbf{W}$, $\mathbf{H}$ and $\boldsymbol{\alpha}$ in the LRTFS model defined by Eqs. (2)-(4). The first approach, described in the next section is maximum joint likelihood estimation (MJLE). It relies on the minimization of $-\log p(\mathbf{x}, \boldsymbol{\alpha}|\mathbf{W}, \mathbf{H}, \lambda)$. The second approach is maximum marginal likelihood estimation (MMLE), described in Section 3.2. It relies on the minimization of $-\log p(\mathbf{x}|\mathbf{W}, \mathbf{H}, \lambda)$, i.e., involves the marginalization of $\boldsymbol{\alpha}$ from the joint likelihood, following the principle of SBL. Though we present MMLE for the sake of completeness, our current implementation does not scale with the dimensions involved in the audio signal processing applications presented in Section 5, and large-scale algorithms for MMLE are left as future work.

## 3.1 Maximum joint likelihood estimation (MJLE)

**Objective.** MJLE relies on the optimization of

$$C_{\text{JL}}(\boldsymbol{\alpha}, \mathbf{W}, \mathbf{H}, \lambda) \overset{\text{def}}{=} -\log p(\mathbf{x}, \boldsymbol{\alpha}|\mathbf{W}, \mathbf{H}, \lambda) \tag{10}$$

$$= \frac{1}{\lambda}\|\mathbf{x} - \boldsymbol{\Phi}\boldsymbol{\alpha}\|_2^2 + D_{\text{IS}}(|\boldsymbol{\alpha}|^2|\mathbf{v}) + \log(|\boldsymbol{\alpha}|^2) + M \log \pi , \tag{11}$$

where we recall that $\mathbf{v}$ is the vectorized version of $\mathbf{WH}$ and where $D_{\text{IS}}(\mathbf{A}|\mathbf{B}) = \sum_{ij} d_{\text{IS}}(a_{ij}|b_{ij})$ is the IS divergence between nonnegative matrices (or vectors, as a special case), with $d_{\text{IS}}(x|y) = (x/y) - \log(x/y) - 1$. The first term in Eq. (11) measures the discrepancy between the raw signal and its approximation. The second term ensures that the synthesis coefficients are approximately low-rank. Unexpectedly, a third term that favors sparse solutions of $\boldsymbol{\alpha}$, thanks to the $\log$ function, naturally appears from the derivation of the joint likelihood. The objective function (11) is not convex and the EM algorithm described next may only ensure convergence to a local solution.

**EM algorithm.** In order to minimize $C_{\text{JL}}$, we employ an EM algorithm based on the architecture proposed by Figueiredo & Nowak [6]. It consists of rewriting Eq. (5) as

$$\mathbf{z} = \boldsymbol{\alpha} + \sqrt{\beta}\,\mathbf{e}_1 , \tag{12}$$

$$\mathbf{x} = \boldsymbol{\Phi}\mathbf{z} + \mathbf{e}_2 , \tag{13}$$

where $\mathbf{z}$ acts as a hidden variable, $\mathbf{e}_1 \sim N_c(\mathbf{0}, \mathbf{I})$, $\mathbf{e}_2 \sim N_c(\mathbf{0}, \lambda\mathbf{I} - \beta\boldsymbol{\Phi}\boldsymbol{\Phi}^*)$, with the operator $\cdot^*$ denoting Hermitian transpose. Provided that $\beta \leq \lambda/\delta_{\boldsymbol{\Phi}}$, where $\delta_{\boldsymbol{\Phi}}$ is the largest eigenvalue of $\boldsymbol{\Phi}\boldsymbol{\Phi}^*$, the likelihood function $p(\mathbf{x}|\boldsymbol{\alpha}, \lambda)$ under Eqs. (12)-(13) is the same as under Eq. (5). Denoting the set of parameters by $\boldsymbol{\theta}_{\text{JL}} = \{\boldsymbol{\alpha}, \mathbf{W}, \mathbf{H}, \lambda\}$, the EM algorithm relies on the iterative minimization of

$$Q(\boldsymbol{\theta}_{\text{JL}}|\tilde{\boldsymbol{\theta}}_{\text{JL}}) = - \int_{\mathbf{z}} \log p(\mathbf{x}, \boldsymbol{\alpha}, \mathbf{z}|\mathbf{W}, \mathbf{H}, \lambda)p(\mathbf{z}|\mathbf{x}, \tilde{\boldsymbol{\theta}}_{\text{JL}})d\mathbf{z} , \tag{14}$$

where $\tilde{\boldsymbol{\theta}}_{\text{JL}}$ acts as the current parameter value. Loosely speaking, the EM algorithm relies on the idea that if $\mathbf{z}$ was known, then the estimation of $\boldsymbol{\alpha}$ and of the other parameters would boil down to the mere white noise denoising problem described by Eq. (12). As $\mathbf{z}$ is not known, the posterior mean value w.r.t $\mathbf{z}$ of the joint likelihood is considered instead.

The complete likelihood in Eq. (14) may be decomposed as

$$\log p(\mathbf{x}, \boldsymbol{\alpha}, \mathbf{z}|\mathbf{W}, \mathbf{H}, \lambda) = \log p(\mathbf{x}|\mathbf{z}, \lambda) + \log p(\mathbf{z}|\boldsymbol{\alpha}) + \log p(\boldsymbol{\alpha}|\mathbf{WH}). \tag{15}$$

The hidden variable posterior simplifies to $p(\mathbf{z}|\mathbf{x}, \boldsymbol{\theta}_{\text{JL}}) = p(\mathbf{z}|\mathbf{x}, \lambda)$. From there, using standard manipulations with Gaussian distributions, the $(i+1)^{th}$ iteration of the resulting algorithm writes as follows.

$$\text{E-step:} \quad \mathbf{z}^{(i)} = \mathbb{E}\{\mathbf{z}|\mathbf{x}, \lambda^{(i)}\} = \boldsymbol{\alpha}^{(i)} + \frac{\beta}{\lambda^{(i)}}\boldsymbol{\Phi}^*(\mathbf{x} - \boldsymbol{\Phi}\boldsymbol{\alpha}^{(i)}) \tag{16}$$

$$\text{M-step:} \quad \forall(f, n),\ \alpha_{fn}^{(i+1)} = \frac{v_{fn}^{(i)}}{v_{fn}^{(i)} + \beta}z_{fn}^{(i)} \tag{17}$$

$$(\mathbf{W}^{(i+1)}, \mathbf{H}^{(i+1)}) = \underset{\mathbf{W}, \mathbf{H} \geq 0}{\arg\min} \sum_{fn} D_{\text{IS}}\left(|\alpha_{fn}^{(i+1)}|^2|[\mathbf{WH}]_{fn}\right) \tag{18}$$

$$\lambda^{(i+1)} = \frac{1}{T}\|\mathbf{x} - \boldsymbol{\Phi}\boldsymbol{\alpha}^{(i+1)}\|_F^2 \tag{19}$$

In Eq. (17), $v_{fn}^{(i)}$ is a shorthand for $[\mathbf{W}^{(i)}\mathbf{H}^{(i)}]_{fn}$ . Eq. (17) is simply the application of Wiener filtering to Eq. (12) with $\mathbf{z} = \mathbf{z}^{(i)}$. Eq. (18) amounts to solving a NMF with the IS divergence; it may be solved using majorization-minimization, resulting in the standard multiplicative update rules given in [3]. A local solution might only be obtained with this approach, but this is still decreasing the negative log-likelihood at every iteration. The update rule for $\lambda$ is not the one that exactly derives from the EM procedure (this one has a more complicated expression), but it still decreases the negative log-likelihood at every iteration as explained in [6].

Note that the overall algorithm is rather computationally friendly as no matrix inversion is required. The $\mathbf{\Phi}\boldsymbol{\alpha}$ and $\mathbf{\Phi}^*\mathbf{x}$ operations in Eq. (16) correspond to analysis and synthesis operations that can be realized efficiently using optimized packages, such as the Large Time-Frequency Analysis Toolbox (LTFAT) [7].

### 3.2 Maximum marginal likelihood estimation (MMLE)

**Objective.** The second estimation method relies on the optimization of

$$C_{\text{ML}}(\mathbf{W}, \mathbf{H}, \lambda) \overset{\text{def}}{=} - \log p(\mathbf{x}|\mathbf{W}, \mathbf{H}, \lambda) \tag{20}$$

$$= - \log \int_{\boldsymbol{\alpha}} p(\mathbf{x}|\boldsymbol{\alpha}, \lambda) p(\boldsymbol{\alpha}|\mathbf{W}\mathbf{H}) d\boldsymbol{\alpha} \tag{21}$$

It corresponds to the "type-II" maximum likelihood procedure employed in [4, 5]. By treating $\boldsymbol{\alpha}$ as a nuisance parameter, the number of parameters involved in the data likelihood is significantly reduced, yielding more robust estimation with fewer local minima in the objective function [5].

**EM algorithm.** In order to minimize $C_{\text{ML}}$, we may use the EM architecture described in [4, 5] that quite naturally uses $\boldsymbol{\alpha}$ has the hidden data. Denoting the set of parameters by $\boldsymbol{\theta}_{\text{ML}} = \{\mathbf{W}, \mathbf{H}, \lambda\}$, the EM algorithm relies on the iterative minimization of

$$Q(\boldsymbol{\theta}_{\text{ML}}|\tilde{\boldsymbol{\theta}}_{\text{ML}}) = - \int_{\boldsymbol{\alpha}} \log p(\mathbf{x}, \boldsymbol{\alpha}|\mathbf{W}, \mathbf{H}, \lambda) p(\boldsymbol{\alpha}|\mathbf{x}, \tilde{\boldsymbol{\theta}}_{\text{ML}}) d\boldsymbol{\alpha}, \tag{22}$$

where $\tilde{\boldsymbol{\theta}}_{\text{ML}}$ acts as the current parameter value. As the derivations closely follow [4, 5], we skip details for brevity. Using rather standard results about Gaussian distributions the $(i+1)^{th}$ iteration of the algorithm writes as follows.

$$\text{E-step}: \quad \mathbf{\Sigma}^{(i)} = (\mathbf{\Phi}^*\mathbf{\Phi}/\lambda^{(i)} + \text{diag}(\mathbf{v}^{(i-1)})^{-1})^{-1} \tag{23}$$

$$\boldsymbol{\alpha}^{(i)} = \mathbf{\Sigma}^{(i)}\mathbf{\Phi}^*\mathbf{x}/\lambda^{(i)} \tag{24}$$

$$\mathbf{v}^{(i)} = \mathbb{E}\{|\boldsymbol{\alpha}|^2|\mathbf{x}, \mathbf{v}^{(i)}, \lambda^{(i)}\} = \text{diag}(\mathbf{\Sigma}^{(i)}) + |\boldsymbol{\alpha}^{(i)}|^2 \tag{25}$$

$$\text{M-step}: \quad (\mathbf{W}^{(i+1)}, \mathbf{H}^{(i+1)}) = \underset{\mathbf{W}, \mathbf{H} \geq 0}{\arg\min} \sum_{fn} D_{\text{IS}}\left(v_{fn}^{(i)}|[\mathbf{W}\mathbf{H}]_{fn}\right) \tag{26}$$

$$\lambda^{(i+1)} = \frac{1}{T}\left[\|\mathbf{x} - \mathbf{\Phi}\boldsymbol{\alpha}^{(i)}\|_2^2 + \lambda^{(i)} \sum_{m=1}^{M}(1 - \mathbf{\Sigma}_{mm}^{(i)}/v_m^{(i)})\right] \tag{27}$$

The complexity of this algorithm can be problematic as it involves the computation of the inverse of a matrix of size $M$ in the expression of $\mathbf{\Sigma}^{(i)}$. $M$ is typically at least twice larger than $T$, the signal length. Using the Woodbury matrix identity, the expression of $\mathbf{\Sigma}^{(i)}$ can be reduced to the inversion of a matrix of size $T$, but this is still too large for most signal processing applications (e.g., 3 min of music sampled at CD quality makes $T$ in the order of $10^6$). As such, we will discard MMLE in the experiments of Section 5 but the methodology presented in this section can be relevant to other problems with smaller dimensions.

## 4   Multi-resolution LRTFS

Besides the advantage of modeling the raw signal itself, and not its STFT, another major strength of LRTFS is that it offers the possibility of multi-resolution modeling. The latter consists of representing a signal as a sum of t-f atoms with different temporal (and thus frequency) resolutions. This is for example relevant in audio where transients, such as the attacks of musical notes, are much shorter than sustained parts such as the tonal components (the steady, harmonic part of musical notes). Another example is speech where different classes of phonemes can have different resolutions. At even higher level, stationarity of female speech holds at shorter resolution than male speech. Because traditional spectral factorizations approaches work on the transformed data, the time resolution is set once for all at feature computation and cannot be adapted during decomposition.

In contrast, LRTFS can accommodate multiple t-f bases in the following way. Assume for simplicity that $\mathbf{x}$ is to be expanded on the union of two frames $\mathbf{\Phi}_a$ and $\mathbf{\Phi}_b$, with common column size $T$

and with t-f grids of sizes $F_a \times N_a$ and $F_b \times N_b$, respectively. $\mathbf{\Phi}_a$ may be for example a Gabor frame with short time resolution and $\mathbf{\Phi}_b$ a Gabor frame with larger resolution – such a setting has been considered in many audio applications, e.g., [8, 9], together with sparse synthesis coefficients models. The multi-resolution LRTFS model becomes

$$\mathbf{x} = \mathbf{\Phi}_a \boldsymbol{\alpha}_a + \mathbf{\Phi}_b \boldsymbol{\alpha}_b + \mathbf{e} \tag{28}$$

with

$$\forall (f,n) \in \{1, \ldots, F_a\} \times \{1, \ldots, N_a\}, \; \alpha_{a,fn} \sim N_c([\mathbf{W}_a \mathbf{H}_a]_{fn}) , \tag{29}$$

$$\forall (f,n) \in \{1, \ldots, F_b\} \times \{1, \ldots, N_b\}, \; \alpha_{b,fn} \sim N_c([\mathbf{W}_b \mathbf{H}_b]_{fn}) , \tag{30}$$

and where $\{\alpha_{a,fn}\}_{fn}$ and $\{\alpha_{b,fn}\}_{fn}$ are the coefficients of $\boldsymbol{\alpha}_a$ and $\boldsymbol{\alpha}_b$, respectively.

By stacking the bases and synthesis coefficients into $\mathbf{\Phi} = [\mathbf{\Phi}_a \, \mathbf{\Phi}_b]$ and $\boldsymbol{\alpha} = [\boldsymbol{\alpha}_a^T \, \boldsymbol{\alpha}_b^T]^T$ and introducing a latent variable $\mathbf{z} = [\mathbf{z}_a^T \, \mathbf{z}_b^T]^T$, the negative joint log-likelihood $-\log p(\mathbf{x}, \boldsymbol{\alpha} | \mathbf{W}_a, \mathbf{H}_a, \mathbf{W}_b, \mathbf{H}_b, \lambda)$ in the multi-resolution LRTFS model can be optimized using the EM algorithm described in Section 3.1. The resulting algorithm at iteration $(i+1)$ writes as follows.

$$\text{E-step:} \quad \text{for } \ell = \{a, b\}, \; \mathbf{z}_\ell^{(i)} = \boldsymbol{\alpha}_\ell^{(i)} + \frac{\beta}{\lambda} \mathbf{\Phi}_\ell^* (\mathbf{x} - \mathbf{\Phi}_a \boldsymbol{\alpha}_a^{(i)} - \mathbf{\Phi}_b \boldsymbol{\alpha}_b^{(i)}) \tag{31}$$

$$\text{M-step:} \quad \text{for } \ell = \{a, b\}, \; \forall (f,n) \in \{1, \ldots, F_\ell\} \times \{1, \ldots, N_\ell\}, \; \alpha_{\ell,fn}^{(i+1)} = \frac{v_{\ell,fn}^{(i)}}{v_{\ell,fn}^{(i)} + \beta} z_{fn}^{(i)} \tag{32}$$

$$\text{for } \ell = \{a, b\}, (\mathbf{W}_\ell^{(i+1)}, \mathbf{H}_\ell^{(i+1)}) = \underset{\mathbf{W}_\ell, \mathbf{H}_\ell \geq 0}{\arg \min} \sum_{fn} D_{\text{IS}} \left( |\alpha_{\ell,fn}^{(i+1)}|^2 | [\mathbf{W}_\ell \mathbf{H}_\ell]_{fn} \right) \tag{33}$$

$$\lambda^{(i+1)} = \|\mathbf{x} - \mathbf{\Phi}_a \boldsymbol{\alpha}_a^{(i+1)} - \mathbf{\Phi}_b \boldsymbol{\alpha}_b^{(i+1)}\|_2^2 / T \tag{34}$$

The complexity of the algorithm remains fully compatible with signal processing applications. Of course, the proposed setting can be extended to more than two bases.

## 5 Experiments

We illustrate the effectiveness of our approach on two experiments. The first one, purely illustrative, decomposes a jazz excerpt into two layers (tonal and transient), plus a residual layer, according to the hybrid/morphological model presented in [8, 10]. The second one is a speech enhancement problem, based on a semi-supervised source separation approach in the spirit of [11]. Even though we provided update rules for $\lambda$ for the sake of completeness, this parameter was not estimated in our experiments, but instead treated as an hyperparameter, like in [5, 6]. Indeed, the estimation of $\lambda$ with all the other parameters free was found to perform poorly in practice, a phenomenon observed with SBL as well.

### 5.1 Hybrid decomposition of music

We consider a 6 s jazz excerpt sampled at 44.1 kHz corrupted with additive white Gaussian noise with 20 dB input Signal to Noise Ratio (SNR). The hybrid model aims to decompose the signal as

$$\boldsymbol{x} = \boldsymbol{x}_{\text{tonal}} + \boldsymbol{x}_{\text{transient}} + \boldsymbol{e} = \mathbf{\Phi}_{\text{tonal}} \boldsymbol{\alpha}_{\text{tonal}} + \mathbf{\Phi}_{\text{transient}} \boldsymbol{\alpha}_{\text{transient}} + \boldsymbol{e} , \tag{35}$$

using the multi-resolution LRTFS method described in Section 4. As already mentionned, a classical design consists of working with Gabor frames. We use a 2048 samples-long ($\sim$ 46 ms) Hann window for the tonal layer, and a 128 samples-long ($\sim$ 3 ms) Hann window for the transient layer, both with a 50% time overlap. The number of latent components in the two layers is set to $K = 3$.

We experimented several values for the hyperparameter $\lambda$ and selected the results leading to best output SNR (about 26 dB). The estimated components are shown at Fig. 1. When listening to the signal components (available in the supplementary material), one can identify the hit-hat in the first and second components of the transient layer, and the bass and piano attacks in the third component. In the tonal layer, one can identify the bass and some piano in the first component, some piano in the second component, and some hit-hat "ring" in the third component.

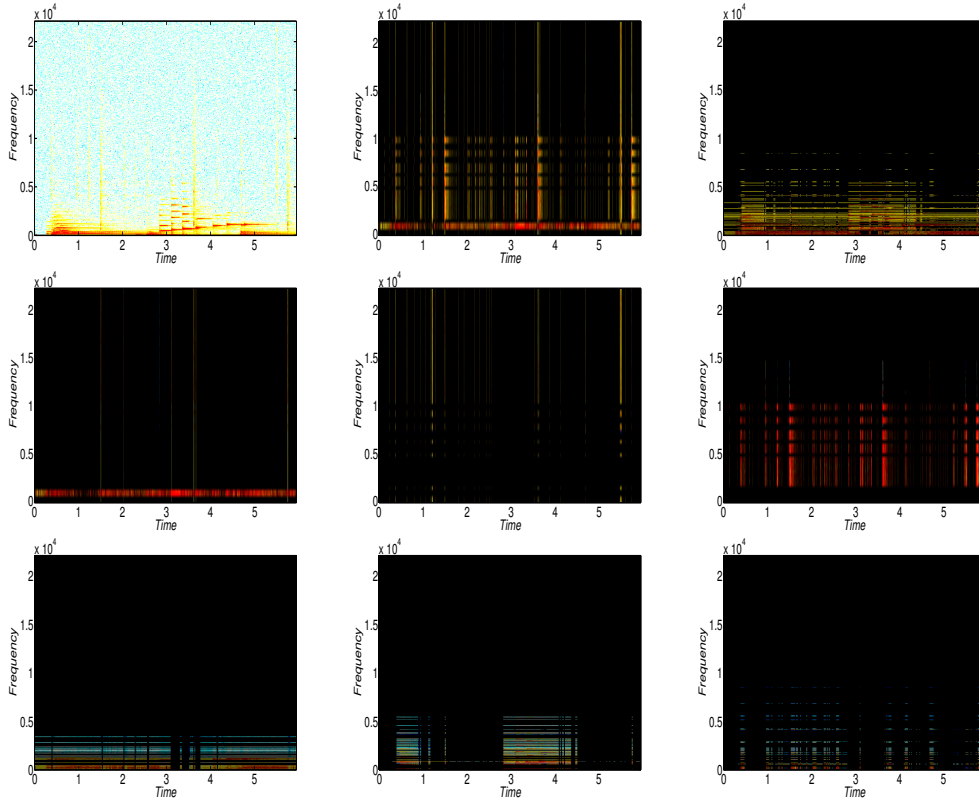

Figure 1: Top: spectrogram of the original signal (left), estimated transient coefficients $\log|\boldsymbol{\alpha}_{\text{transient}}|$ (center), estimated tonal coefficients $\log|\boldsymbol{\alpha}_{\text{tonal}}|$ (right). Middle: the 3 latent components (of rank 1) from the transient layer. Bottom: the 3 latent components (of rank 1) from the tonal layer.

## 5.2 Speech enhancement

The second experiment considers a semi-supervised speech enhancement example (treated as a single-channel source separation problem). The goal is to recover a speech signal corrupted by a texture sound, namely applauses. The synthesis model considered is given by

$$\boldsymbol{x} = \boldsymbol{\Phi}_{\text{tonal}}\left(\boldsymbol{\alpha}_{\text{tonal}}^{\text{speech}} + \boldsymbol{\alpha}_{\text{tonal}}^{\text{noise}}\right) + \boldsymbol{\Phi}_{\text{transient}}\left(\boldsymbol{\alpha}_{\text{transient}}^{\text{speech}} + \boldsymbol{\alpha}_{\text{transient}}^{\text{noise}}\right) + \boldsymbol{e}, \tag{36}$$

with

$$\boldsymbol{\alpha}_{\text{tonal}}^{\text{speech}} \sim N_c\left(\boldsymbol{0}, \boldsymbol{W}_{\text{tonal}}^{\text{train}}\boldsymbol{H}_{\text{tonal}}^{\text{speech}}\right), \quad \boldsymbol{\alpha}_{\text{tonal}}^{\text{noise}} \sim N_c\left(\boldsymbol{0}, \boldsymbol{W}_{\text{tonal}}^{\text{noise}}\boldsymbol{H}_{\text{tonal}}^{\text{noise}}\right), \tag{37}$$

and

$$\boldsymbol{\alpha}_{\text{transient}}^{\text{speech}} \sim N_c\left(\boldsymbol{0}, \boldsymbol{W}_{\text{transient}}^{\text{train}}\boldsymbol{H}_{\text{transient}}^{\text{speech}}\right), \quad \boldsymbol{\alpha}_{\text{transient}}^{\text{noise}} \sim N_c\left(\boldsymbol{0}, \boldsymbol{W}_{\text{transient}}^{\text{noise}}\boldsymbol{H}_{\text{transient}}^{\text{noise}}\right). \tag{38}$$

$\boldsymbol{W}_{\text{tonal}}^{\text{train}}$ and $\boldsymbol{W}_{\text{transient}}^{\text{train}}$ are fixed pre-trained dictionaries of dimension $K = 500$, obtained from 30 min of training speech containing male and female speakers. The training data, with sampling rate 16kHz, is extracted from the TIMIT database [12]. The noise dictionaries $\boldsymbol{W}_{\text{tonal}}^{\text{noise}}$ and $\boldsymbol{W}_{\text{transient}}^{\text{noise}}$ are learnt from the noisy data, using $K = 2$. The two t-f bases are Gabor frames with Hann window of length 512 samples ($\sim 32\,\text{ms}$) for the tonal layer and 32 samples ($\sim 2\,\text{ms}$) for the transient layer, both with 50% overlap. The hyperparameter $\lambda$ is gradually decreased to a negligible value during iterations (resulting in a negligible residual $\boldsymbol{e}$), a form of warm-restart strategy [13].

We considered 10 test signals composed of 10 different speech excerpts (from the TIMIT dataset as well, among excerpts not used for training) mixed in the middle of a 7 s-long applause sample. For every test signal, the estimated speech signal is computed as

$$\hat{\boldsymbol{x}} = \boldsymbol{\Phi}_{\text{tonal}}\hat{\boldsymbol{\alpha}}_{\text{tonal}}^{\text{speech}} + \boldsymbol{\Phi}_{\text{transient}}\hat{\boldsymbol{\alpha}}_{\text{transient}}^{\text{speech}} \tag{39}$$

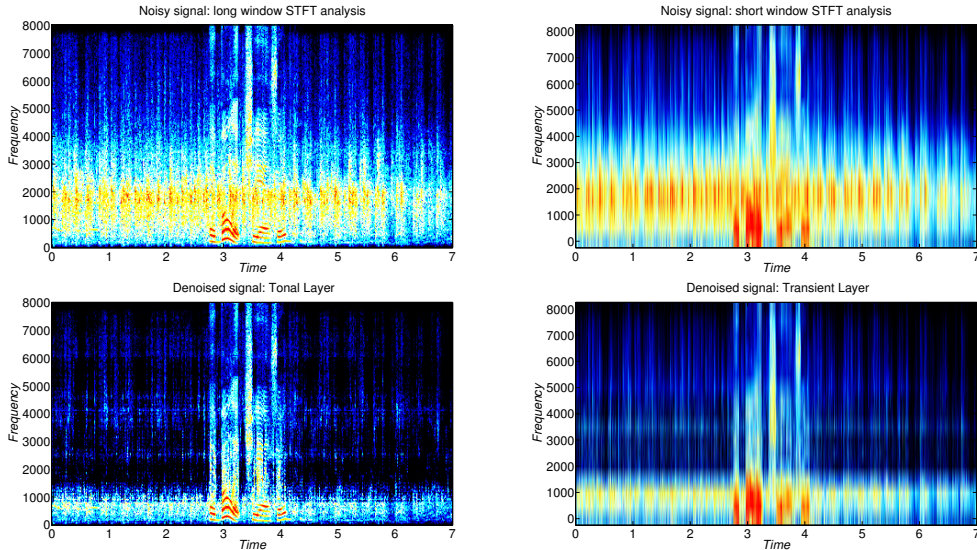

Figure 2: Time-frequency representations of the noisy data (top) and of the estimated tonal and transient layers from the speech (bottom).

and a SNR improvement is computed as the difference between the output and input SNRs. With our approach, the average SNR improvement other the 10 test signals was $6.6$ dB. Fig. 2 displays the spectrograms of one noisy test signal with short and long windows, and the clean speech synthesis coefficients estimated in the two layers. As a baseline, we applied IS-NMF in a similar setting using one Gabor transform with a window of intermediate length (256 samples, $\sim 16$ ms). The average SNR improvement was $6$ dB in that case. We also applied the standard OMLSA speech enhancement method [14] (using the implementation available from the author with default parameters) and the average SNR improvement was $4.6$ dB with this approach. Other experiments with other noise types (such as helicopter and train sounds) gave similar trends of results. Sound examples are provided in the supplementary material.

## 6  Conclusion

We have presented a new model that bridges the gap between t-f synthesis and traditional NMF approaches. The proposed algorithm for maximum joint likelihood estimation of the synthesis co-efficients and their low-rank variance can be viewed as an iterative shrinkage algorithm with an additional Itakura-Saito NMF penalty term. In [15], Elad explains in the context of sparse representations that soft thresholding of analysis coefficients corresponds to the first iteration of the forward-backward algorithm for LASSO/basis pursuit denoising. Similarly, Itakura-Saito NMF followed by Wiener filtering correspond to the first iteration of the proposed EM algorithm for MJLE.

As opposed to traditional NMF, LRTFS accommodates multi-resolution representations very naturally, with no extra difficulty at the estimation level. The model can be extended in a straightforward manner to various additional penalties on the matrices $\mathbf{W}$ or $\mathbf{H}$ (such as smoothness or sparsity). Future work will include the design of a scalable algorithm for MMLE, using for example message passing [16], and a comparison of MJLE and MMLE for LRTFS. Moreover, our generative model can be considered for more general inverse problems such as multichannel audio source separation [17]. More extensive experimental studies are planned in this direction.

#### Acknowledgments

The authors are grateful to the organizers of the *Modern Methods of Time-Frequency Analysis Semester* held at the Erwin Schröedinger Institute in Vienna in December 2012, for arranging a very stimulating event where the presented work was initiated.

## Footnotes

[2]In the general unsupervised setting where both $\mathbf{W}$ and $\mathbf{H}$ are estimated, $\mathbf{WH}$ must be low-rank such that $K < F$ and $K < N$. However, in supervised settings where $\mathbf{W}$ is known, we may have $K > F$.

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
