[Reviews · NeurIPS 2014]

Submitted by Assigned_Reviewer_6

This paper presents a extension in the family of low-rank spectrogram-factorization models of sound decomposition that generalizes the latent factors over an arbitrary (potentially overcomplete) set of bases, meaning that decomposition can be performed on "multiresolution" (e.g. tonal + transient) bases. This is argued to help with source separation and denoising. It starts from Gaussian Composite Model of Fevotte et al, which models the variance of each cell of a spectrogram with a low-rank matrix factorization, but then expands the implicit convolution with the multiple gabor impulse responses corresponding to each row of the spectrogram to give a purely time-domain expression for what is being estimated, that can then be modified to work for different sets of impulse responses, including nonorthogonal ones. An EM (local optimum) analysis procedure is presented.

Quality: I think this is an interesting insight, although I'm still trying to figure out the exact implications. It seems like, at the outset, it's just another way of understanding Fevotte's GCM, but the ability to include overcomplete dictionaries is nice. On the other hand, it seems like this ought to cause some complications, and I didn't immediately understand why these didn't cause more trouble. I assume the estimation procedure is correct (and I'm impressed by this kind of analysis) although I couldn't follow it in detail.

Clarity: The work is well explained and motivated. The evaluation, however, was very superficial - we're given one purely illustrative result, and one SNR improvement measurement: I hardly think we can draw any conclusions from a single example. I was grateful for the included sound examples, but I wasn't all that impressed. For one thing, the OMLSA enhanced example seems to be much longer (7 sec c/w 2 sec) than the new result, which makes me wonder if the comparison has been done fairly. Secondly, although OMLSA includes more background noise (in part because it is longer), the speech sounds much higher quality to me. The denoising of the music example comes at the expense of very significant artifacts - I would expect conventional wavelet shrinkage or weiner filtering to give much higher perceived quality in this case.

Originality: The work seems to rely very heavily on the Fevotte et al foundation, although the time-domain interpretation certainly made me think about it in a new way.

Significance: On the face of it, the ability to apply matrix-factorization techniques to overcomplete bases seems like it could have significant impact. However, the examples and evaluation presented in this work were too thin and superficial to really show any of this, so as it stands it is an act of faith only to rate this work as having substantial significance.
Summary: Reformulation of probabilistic matrix-factorization sound analysis in terms of time-domain kernels allows an EM analysis for multiresolution (overcomplete) bases. However, thin evaluation and significant artifacts in examples fail to demonstrate its importance in the current form.

Submitted by Assigned_Reviewer_16

While the regular NMF framework considers a generative model of the TF analysis coefficients of a signal, the proposed framework considers a generative model of the time-domain signal itself by describing a generative model of TF synthesis coefficients. This allows us to open doors to multi-resolution representations, that were not possible in the traditional NMF setting.

This is an interesting paper. In particular, I personally like the idea of the multi-resolution extension a lot.

While the authors present some results on multi-resolution signal decompositions, I am also curious to know the performance of the proposed method in the single-resolution case, compared with IS-NMF. Does it give similar W and H? Is the present algorithm as fast and stable as the multiplicative update algorithm for IS-NMF?
Summary: While the regular NMF framework considers a generative model of the TF analysis coefficients of a signal, the proposed framework considers a generative model of the time-domain signal itself by describing a generative model of TF synthesis coefficients. This allows us to open doors to multi-resolution representations, that were not possible in the traditional NMF setting.

Submitted by Assigned_Reviewer_23

This paper presents a thoughtful interpretation of time/frequency factorizations. Instead of using existing formulations that are build on the analysis coefficients of STFT transforms, it presents a more flexible framework that can be applied on an arbitrary transform and is directly linked to the time domain signal.

The paper is easy to read for people in this area and of good quality. The originality is above average, although the significance is not fully communicated in the paper. Perhaps the most disappointing part of this paper is in terms of the results and the evaluation. The supplemental material is not particularly convincing and the paper itself doesn't compare this work to the methods it builds on. I don't think that this is a significantly weak point (the theoretical development is itself enough to make this an interesting paper), but I think it would have made the paper a lot better.
Summary: Good paper, well written, solid theory. I would have liked to see more results and comparisons, but this isn't a fatal flaw in this paper since the theoretical contribution is very interesting.
Author Feedback
Author rebuttal: We would like to thank the reviewers for their constructive comments. A concern shared by the reviewers seems to be about the experiments. In that respect, we would like to stress that, from our point of view, the main contribution of this paper is the proposed synthesis model built on the GCM, with its estimation. Our contribution pushes the limits of NMF for signal representation and we hope this is valuable in itself. Because of limited space, we made the choice of writing a methodological paper with two simple examples illustrating the potentials of LRTFS. It was difficult to address a more complete experimental analysis in the 2 pages we dedicated to experiments (and many technical details were already left apart in the methodological part). So we agree about the comments made on the experimental part, but we plan to address a more detailed analysis in a longer paper. However, in the final version, we will try to add a short comparison with standard IS-NMF as rightly suggested by one of the reviewers.

Answers to specific concerns: 

*Assigned_Reviewer_16

-"[...] Does it give similar W and H?". This is a good question. We hesitated to write a pure qualitative experimental section, instead of the speech enhancement experiments. However, showing various W and H is very space consuming, and always gives very subjective interpretations. It would be more valuable to discuss on the W and H matrices on an application such as pitch tracking, where these matrices hold the information of interest. This could be included in a longer version.

-"Is the present algorithm as fast and stable as the multiplicative update algorithm for IS-NMF?" Every step of MMLE is actually akin to IS-NMF on the posterior synthesis coefficients, so our algorithm is bound to have higher complexity than IS-NMF. Stability is more difficult to evaluate, but we did not observe any pathological behavior. 

*Assigned_Reviewer_23

- "the paper itself doesn't compare this work to the methods it builds on". This is a right comment and we will attempt to add a very short comparison with IS-NMF in the single resolution case in the final version.

*Assigned_Reviewer_6

-"It seems like, at the outset, it's just another way of understanding Fevotte's GCM". Fevotte's GCM was our starting point to write the model. We propose a strong generalization of it, as the GCM can be viewed as a new form of prior on the synthesis coefficients, which can then be estimated in a Bayesian procedure. This is very novel (as far as we are aware).

-"OMLSA enhanced example seems to be much longer (7 sec c/w 2 sec) than the new result". This is a little mistake made during the creation of the sound files. For the results of LRTFS, we only provide the speech without the noise before and after. While for the OMLSA, we provide all the soundtrack, with the noise before and after the speech. However, the noisy speech input is the same for the two algorithms. This will be corrected in the sound files.

-"The denoising of the music example comes at the expense of very significant artifacts".  We prefer the results given by our proposed method, but this is rather subjective (and we might be involuntarily biased toward our own method...). We agree that our method can surely be improved for speech enhancement. However, speech enhancement is only an illustration and was not at all the main focus of the paper. So we agree with the reviewers' remarks concerning this part, but again, we wanted to focus on methodology in this initial contribution to LRTFS.